# The Development of an Ultra-Performance Liquid Chromatography–Tandem Mass Spectrometry Method for Biogenic Amines in Fish Samples

**DOI:** 10.3390/molecules28010184

**Published:** 2022-12-26

**Authors:** Tong Li, Ruiguo Wang, Peilong Wang

**Affiliations:** Institute of Quality Standards and Testing Technology for Agro-Products, Chinese Academy of Agricultural Sciences, Beijing 100081, China

**Keywords:** biogenic amines, benzoyl chloride, derivatization, UPLC–MS/MS

## Abstract

Biogenic amines (BAs) are a group of substances that are formed from amino acids by decarboxylation or amination and transamination of aldehydes and ketones. They may have either an aliphatic, aromatic, or heterocyclic structure. Their quantity determines their effects and optimum amounts are essential for physiological functions, but excess BAs causes various toxic effects throughout the human body. In our study, to rapidly determine 14 BAs (histamine, tyramine, dopamine, tryptamine, serotonin, putrescine, spermine, spermidine, octopamine, benzylamine, 1-Phenylethanamine, cadaverine, 2-Phenethylamine, and agmatine) in real fish samples, an ultra-performance liquid chromatography–tandem mass spectrometry method was established. The fish sample was extracted by acetonitrile with 0.1% formic acid and stable biogenic amine derivatives could be obtained by benzoyl chloride derivatization with a shorter reaction time. The method showed good linearity with a linear range of 3–4 orders of magnitude and regression coefficients ranging from 0.9961 to 0.9999. The calculated LODs ranged from 0.1 to 20 nM and the LOQs ranged from 0.3 to 60 nM. Satisfactory recovery was obtained from 84.6% to 119.3%. The proposed method was employed to determine the concentration levels of biogenic amine derivatives in different fish. The results indicated that this method was suitable for the analysis of biogenic amines.

## 1. Introduction

Biogenic amines are widely found in organisms and a variety of foods, which are a kind of polar or semi-polar low molecular organic matter containing nitrogen. According to their molecular structure, they can be divided into aliphatic amines, aromatic amines, and heterocyclic amines. BAs in food are usually produced by corresponding amino acids by decarboxylase from microorganisms [1]. They exist in animals and plants, fermented foods, meat products, and aquatic products [2]. BAs are an indispensable part of biologically active cells and have important biological functions. However, if the intake of BAs is beyond the body’s burden, some adverse reactions will be caused, including headache, nausea, allergic reaction fever, rash, vomiting, and other toxic reactions. It is well-known that high concentrations of histamine in aquatic products can cause allergies in humans, and such allergic poisonings occur frequently. Therefore, many countries have established limits for histamine content in aquatic products, but limits for other BAs are scarce. For example, the U.S. Food and Drug Administration (FDA) requires that the histamine content in aquatic products should be no higher than 50 mg/kg. The European Union (EU) stipulates that the histamine content should be less than 100 mg/kg. The Chinese Hygiene Standards of Aquatic Products requires that the limit standard of histamine in aquatic products should be no higher than 300 mg/kg [3]. Therefore, BA content in food can be utilized as an indicator of food quality. Therefore, a rapid, reliable, and efficient analytical method for detecting BAs in food are of great significance.

Various analytical methods have been applied in the detection of BAs, such as thin-layer chromatography (TLC) capillary electrophoresis (CE), gas chromatography (GC), polymerase chain reaction (PCR), and liquid chromatography (LC) [4,5,6,7,8]. PCR, as a common method in molecular biology, was simple, rapid, and accurate, which could detect the gene encoding the production of amino acid decarboxylase by using special primers. However, interference of the complex matrix hampered the wide use of this method. LC, especially ultra-performance liquid chromatography (UPLC), has become an important method for the detection of BAs due to its high sensitivity and resolution. The most commonly used detectors include ultraviolet (UV) and fluorescence (FL) detectors. In recent years, UPLC-tandem mass spectrometry (UPLC–MS/MS) equipped with an electrospray ionization (ESI) source is one of the most popular surveillance tools because of its higher selectivity and sensitivity compared with liquid chromatography with a fluorescence detector (LC–FLD) and gas chromatography–mass spectrometry (GC–MS) [9,10,11].

Derivatization is often employed to improve BA response in LC–UV, FL, or MS analyses. For UV and fluorescence detection, derivatization can help BAs to be more easily detected. For mass spectrometry, derivatization helps to increase retention and improve ionization efficiency and sensitivity. Then, the most commonly used derivative reagents are dansyl chloride (Dns-Cl), o-phthalaldehyde (OPA), and benzoyl chloride, in that biogenic amines usually contain amino groups. Dns-Cl derivatization usually requires heating conditions and long reaction times, whereas OPA reacts only with primary amines and the derivatives are unstable. In contrast, benzoyl chloride derivatization is simpler, requiring only a short reaction time under mild reaction conditions [12,13].

In our study, a sensitive and efficient analysis method was developed to determine the 14 BAs (histamine, tyramine, dopamine, tryptamine, serotonin, putrescine, spermine, spermidine, octopamine, benzylamine, 1-Phenylethanamine, cadaverine, 2-Phenethylamine, and agmatine, Figure 1) content in fish, combine with acid extraction and derivatization. Formic acid as an extraction reagent was friendly to the mass spectrometer and commonly used in many laboratories. Benzoyl chloride was used as a derivatization reagent to make the derivatization reaction more rapid and simple. In addition, based on the established method, two types of fish from different markets were analyzed.

## 2. Results and Discussion

### 2.1. Optimization of the LC–MS/MS Conditions

The main MS/MS parameters, including ion source mode, MRM transitions, and their collision energies, were optimized by infusing 1000 nM standard solution of the individual BAs at 10 μL/min directly into the mass spectrometer. For the analysis of BA derivatives, the positive and negative ESI modes were evaluated. The results showed that BA derivatives could not be detected in negative ESI mode, as former studies reported [14,15,16,17]. Hence, the positive ionization mode was employed for the detection of BA derivatives. Normally, each precursor ion produces two product ions as ion transitions. The product ion with the higher response is used for quantitative analysis and another product ion was for auxiliary qualitative analysis. The optimal collision energy (CE) was optimized for each product ion presented in Table 1.

As for LC analysis, the Waters HSS T3 column was chosen as the separation column. A total of 0.1% formic acid and 1 mM ammonium formate in water and acetonitrile were the mobile phases A and B, respectively. In order to obtain a good chromatographic separation effect, mobile phase elution gradient and flow rate were optimized. It could be found that a flow rate of 0.3 mL/min with gradient 0–1 min, 2% B; 1–4.5 min, from 2% to 45% B; 4.5–12 min, from 45% to 98% B; 12–13 min, 98% B; 13–13.1 min, from 98% to 2% B and; 13.1–15 min, 2% B had the best separation effect. The run time and column temperature were finally set to 15 min and 40 °C, respectively. The separation chromatogram of standard BA derivatives is shown in Figure 2. All the 14 derivatized BAs in standard solution were detected under the optimized instrumental conditions. The retention times (RT) of biogenic amine derivatives are displayed in Table 1. A chromatogram of 14 derivatized BAs in the spiked fish sample (1000 nM) was shown in Appendix A.

### 2.2. Optimization of the Sample Preparation

Acid solvents, such as trichloroacetic acid (TCA), hypochlorous acid (HClO), hydrochloric acid (HCl), etc., were mostly used as extraction reagents of BAs from food samples in the previous studies [18,19,20]. When mass spectrometry was used as a detector, these chlorides that were retained in ion channels could produce intense ion suppression in negative detection mode. A recent study showed that satisfactory recovery rates were obtained using SSA as the extraction reagent by comparing the extraction effects between TCA and sulfosalicylic acid (SSA) on polyamines from milk, which suggested organic acid had the potential to be utilized as the extract of BAs [21]. We optimized the extraction reagent by comparing the recoveries of BA derivatives in the spiked samples (1000 nM) extracted with formic acid and acetic acid, respectively. Acetonitrile with 0.1%/0.5% formic acid and acetonitrile with 0.1%/0.5% acetic acid were evaluated. Appendix A demonstrated that recoveries were satisfactory when the extraction reagent was acetonitrile with 0.1% formic acid. However, more acetic acid (0.5%) in acetonitrile reached equal extraction efficiencies. Hence, formic acid was selected as an extraction reagent in our study, which was accessible in the lab and friendly to the mass spectrometer. In addition, according to the pretreatment method of fish tissue samples in the analysis of chemical pollutants, fish samples were pretreated by homogenization and sonication for the full extraction of BAs [22,23]. The extraction method greatly reduces the analysis time, the number of samples and extraction reagents is also greatly reduced, and is friendly to the environment. Finally, the method of extraction of BAs from fish was established by using homogenization and sonication with formic acid as the extraction reagent.

### 2.3. Optimization of the Derivation Conditions

Despite the inherent advantages of LC–MS/MS without any derivatization, matrix component effects should not be ignored, which can affect the ionization of analytes and their response due to signal suppression or enhancement caused by co-eluted compounds. Due to this reason, and also to reduce the polarity of BAs for separation on normal reversed-phase C18 columns, LC–MS/MS methods also utilize derivatization reactions [24,25]. Acyl chlorides were reported as the most dominant derivatization reagents for the analysis of Bas, and we compared three reported derivatization reagents: benzoyl chloride, dansyl chloride, and pyridine-3-sulfonyl chloride. The UPLC conditions and MRM acquisition parameters for three categories of derivatives of all 14 BAs were optimized by a derived standard solution (1000 nM). As shown in Appendix A, sensitivities of BAs derived with benzoyl chloride were higher than that of BAs derived with dansyl chloride and pyridine-3-sulfonyl chloride (1–150 times). Therefore, benzoyl chloride was the optimal derivatization reagent, which exhibited excellent sensitivity for the analysis of BAs. To obtain an effective derivation, the concentration of benzoyl chloride and derivation time (vortex time) has been optimized, respectively. To optimize the concentration of benzoyl chloride, the four concentration levels of benzoyl chloride in ACN (comprising 1%, 2%, 3%, 4%, and 10%, volume ratio) were prepared and employed for the derivation of BAs. The optimization results are shown in Figure 3A. It was obvious when the concentration of benzoyl chloride was 2% and the derivatization achieved maximum efficiency. It was obvious when the concentration of benzoyl chloride was 2% that the derivatization achieved maximum efficiency. Then, the derivatization efficiency reduced extremely with the increase in benzoyl chloride. The possible reason was due to the excess unreacted benzoyl chloride could not be removed completely during the pretreatment after the derivatization reaction and the high content of benzoyl chloride, as residuals of the derivatization reaction would generate ionization competition against biogenic amine derivatives in the ion source. So, 2% Benzoyl chloride in acetonitrile was determined as the optimal concentration for BAs derivatization. In addition, four timepoints (5, 10, 20, and 30 min) were selected to optimize the derivatization reaction time. As shown in Figure 3B, 20 min was enough for the total accomplishment of the derivatization reaction. After comprehensive consideration, 20 min was chosen as the optimal derivatization time.

### 2.4. Validation of the Proposed Method

In order to evaluate the method’s reliability, an investigation of linearity, the limit of detection (LOD), the limit of quantitation (LOQ), accuracy, precision, and the matrix effect was performed. The analytical performance parameters of this optimized LC–MS/MS method for the BA derivatives are shown in Table 2. For linearity, LOD, and LOQ analysis, twenty-one concentration levels (see Material and Methods) were prepared. After derivatization and analysis, linear calibration curves, based on internal standard corrected response versus concentrations, were obtained in the wide linear range (3–4 orders of magnitude) with high correlation coefficients (R2) ranging from 0.9961 to 0.9999. The wider quantification range, compared with former studies, can satisfy the quantification of samples with different concentrations [18,26]. The calculated LODs ranged from 0.13 to 19.90 nM and the LOQs ranged from 0.44 to 65.67 nM. Compared with previous methods reported for the determination of different BA derivatives in biological samples, the present method showed about a 50–200-fold lower LOD than BAs derived by diethyl ethoxymethylenemalonate, a 10–20-fold lower LOD than BAs derived by dansyl chloride [27,28], which indicated the reasonable sensitivity was obtained in our study. As shown in Table 2, matrix effects were almost negative values. They illustrated the suppressions of the analyte signal. The matrix effect mainly resulted from the co-extracted matrix components during the sample extraction procedure. In our study, the conclusion of having no matrix effect would generally be drawn, owing to its value ranging from −20% to 20%. The results suggested the sample preparation procedure in this study showed extraction selectivity of BAs in fish samples. The BA standards were spiked into the blank matrix samples to obtain the spiked samples. Accuracy and precision were expressed as recoveries and relative standard deviations (RSDs) of spiked samples at three concentration levels (10 nM, 50 nM, and 1000 nM). Six replicates of each concentration level were performed. After extraction, derivatization, and LC–MS/MS analysis, recoveries of 14 BAs at low, medium, and high concentration levels ranged from 90.5% to 119.3%, from 84.6% to 115.4%, and from 74.9% to 108.8%, respectively (Table 3). Most RSDs of recovery at three concentration levels were below 10%. These results indicated that this method could obtain reasonable accuracy and precision, compared with the existing method in the literature [29].

### 2.5. Applications

The method was further applied to the analysis of 10 mackerel and 10 tuna samples. Half of the total was purchased from the morning market and the other half was bought from a supermarket in Beijing, respectively. Samples were packed separately in sealing bags with ice and transported to the laboratory immediately. Edible parts of samples were cut and ground to be homogeneous for detection. To detect the content of BAs in fresh fishes, the fish were pretreated and derivatized for LC–MS analysis immediately after the purchase. Table 4 gives information on BA content in two kinds of fish from a morning market and a supermarket, respectively. It could be found that mackerel had much higher histamine, tyramine, cadaverine, and 2-phenethylamine than that in tuna, which could cause a bad smell. Relatively higher histamine, tyramine in mackerel, and putrescine in tuna purchased from a supermarket may result from long storage time, which was consistent with the results of former studies [30].

## 3. Materials and Methods

### 3.1. Chemicals and Materials

Histamine, cadaverine, putrescine dihydrochloride, tyramine, spermine, dopamine, tryptamine, serotonin, spermidine, 2-phenethylamine, octopamine, benzylamine, 1-phenylethanamine, agmatine were purchased from Sigma-Aldrich (St. Louis, Mo, USA). Methanol (MetOH, HPLC grade) and Acetonitrile (ACN, HPLC grade) was purchased from Fisher Chemicals (Bridgewater, NJ, USA). Benzoyl chloride (≥99%), Benzoyl chloride-d5 (≥99%), and formic acid (FA) were bought from Sigma Aldrich (St. Louis, Mo, USA). Distilled water was prepared by a Milli-Q Synthesis water purification system (Millipore, Bedford, MA, USA)

### 3.2. Derivatization

In our study, the BAs were derivatized by benzoyl chloride. The usage of benzoyl chloride and the derivatization time were optimized to obtain the best reaction effect. In our study, BAs derivatized with benzoyl chloride served as normal standards and BAs derivatized with benzoyl chloride-d5 worked as internal standards. The derivatization scheme is shown in Figure 4. After optimization, the stock normal standards and stock internal standards mixture of the corresponding concentration were prepared, respectively. An aliquot of 10 µL of the BAs mixture was diluted with 70 µL 80% acetonitrile containing 0.1% formic acid, then derivatized for 20 min at room temperature after the addition of 40 µL 100 mmol/L sodium carbonate solution and 40 µL 2% benzoyl chloride (or benzoyl chloride-d5) acetonitrile solution in order. An aliquot of 200 µL of the stock internal standard was stabilized with 3800 µL of 65% acetonitrile (0.1% formic acid), which was marked as the internal standard for correction analysis. To prepare the calibration standards, twelve different amounts of normal standards and fixed amounts of internal standards were mixed for twelve points of the calibration curve.

### 3.3. Sample Information and Sample Preparation

Two kinds of fish, including mackerel and tuna, were purchased from a morning market and a supermarket in Beijing, respectively. Samples were packed separately in sealing bags with ice and transported to the laboratory as soon as possible. The bones, skins, and internal organs of these fish were removed immediately, and then the fish tissue was homogenized roughly in a blender (JXFSTPRP-24, Jingxin Technologies, Shanghai, China). One part of each fish tissue was analyzed at once, and the remaining tissues were divided into several parts and stored at −20 °C for further analysis.

An aliquot of each individual sample (~20 mg) was precisely weighed and transferred to an Eppendorf tube. After the addition of 80 μL of extraction solvent (precooled at −20 °C and acetonitrile with 0.1% formic acid) and 20 μL of H_2_O, the samples were vortexed for 30 s, homogenized for 4 min, and sonicated for 5 min in an ice-water bath. The homogenate and sonicate circle were repeated three times, followed by subsiding at −40 °C overnight and centrifuging at 12,000 rpm and 4 °C for 15 min. An 80 µL of the supernatants were transferred to the centrifuge tube and incubated for 20 min after adding 40 µL 100 mM sodium carbonate solution and 40 µL 2% benzoyl chloride acetonitrile solution. The samples were centrifuged at 12,000 rpm for 15 min at 4 °C after the addition of a 10 µL internal standard. The 40 μL of the supernatants were added to 20 µL H_2_O and then transferred to an auto-sampler vial for UPLC–MS/MS analysis.

### 3.4. UPLC–MS/MS Analysis

A SCIEX ExionLC system coupled with an AB Sciex QTrap 6500+ mass spectrometer (AB Sciex, Framingham, MA, USA) and equipped with an electrospray ionization (ESI) source was used in the BA analysis. All of the standards and samples were separated on a Waters ACQUITY UPLC HSS T3 (100 × 2.1 mm, 1.8 μm, 100Å). The temperature of the automatic injector was 15 °C. The column was maintained at 40 °C and the injection volume was 2 μL. The flow rate was 0.3 mL min^−1^. The mobile phases consisting of 0.1% formic acid and 1 mM ammonium formate in water (A) and acetonitrile (B) were used with gradient elution. The initial conditions were 2% B for 1 min, ramped to 45% for 4.5 min, ramped to 98% for 12 min, held for 1 min to 13 min, and returned to the initial conditions for 13.1 min, which were then equilibrated for 1.9 min before the injection of the next sample.

The MS analysis was in multiple-reaction monitoring (MRM) mode and the ESI source was operated in positive ion mode for biogenic amines derivatives detection. The optimized parameters were as follows: a source capillary voltage of +5000 V, a declustering potential of65 V, a curtain gas of 35 psi, a gasification temperature of 400 °C, an ion source gas of 1:60 psi, an ion source gas of 2:60 psi, and a collision gas of medium. The MS parameters of retention time, optimized ion transitions, and collision energies are listed in Table 1. Instrument control and MRM data processing were performed using Analyst 1.63 and Multiquant 3.03 software.

### 3.5. Method Validation

The proposed method was evaluated by accuracy, precision, linearity, the limit of detection (LOD), the limit of quantitation (LOQ), and the matrix effect. Twelve concentration levels and fixed concentrations of the internal standards were designed for the examination of linearity. These curves were performed according to the ratios of the peak areas of the normal standards to those of the internal standards plotted against the concentration of the normal standard. LOD and LOQ were calculated based on the S/N (ratio of signal to noise) = to 3 and 10, respectively, based on the sample at the lowest spiked concentration level. Accuracy and precision were expressed as recoveries and relative standard deviations (RSDs), which were evaluated by three concentration levels (10 nM, 50 nM, and 1000 nM) spiked in six replicates of blank fish samples. Matrix effects were evaluated by the following equation:Matrix effect (%) = [(B/A) − 1] × 100%(1)
where A represents the peak area of the standard solution and B represents the peak area of fish sample extract spiked at the same concentration of the standard before LC–MS analysis. Considering the individual differences, six lots of fish were used to evaluate the matrix effect.

## 4. Conclusions

In summary, a sensitive and efficient method of UPLC–MS/MS was established for the simultaneous determination of 14 BAs, combined with acid extraction and derivatization. Formic acid as an extraction reagent was friendly to the mass spectrometer and is cost-effective. Benzoyl chloride was used as a derivatization reagent to make the derivatization reaction more rapid and simple. In addition, MRM mode for BA detection possesses fine accuracy, precision, and wide linearity. Consequently, this analytical method will enable researchers to detect BAs in fish samples successfully and comprehensively.

## Figures and Tables

**Figure 1 molecules-28-00184-f001:**
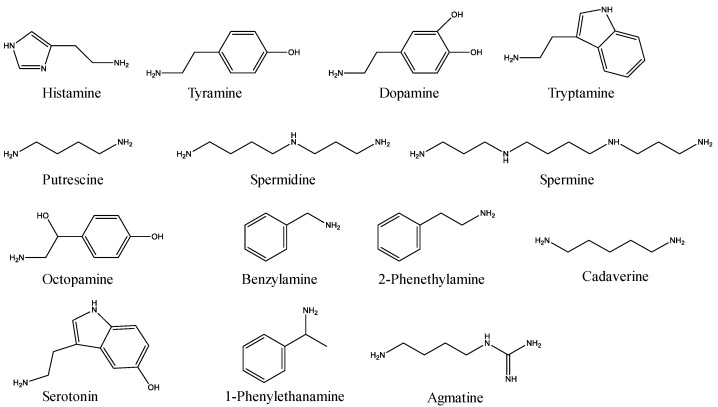
Chemical structures of the 14 BAs.

**Figure 2 molecules-28-00184-f002:**
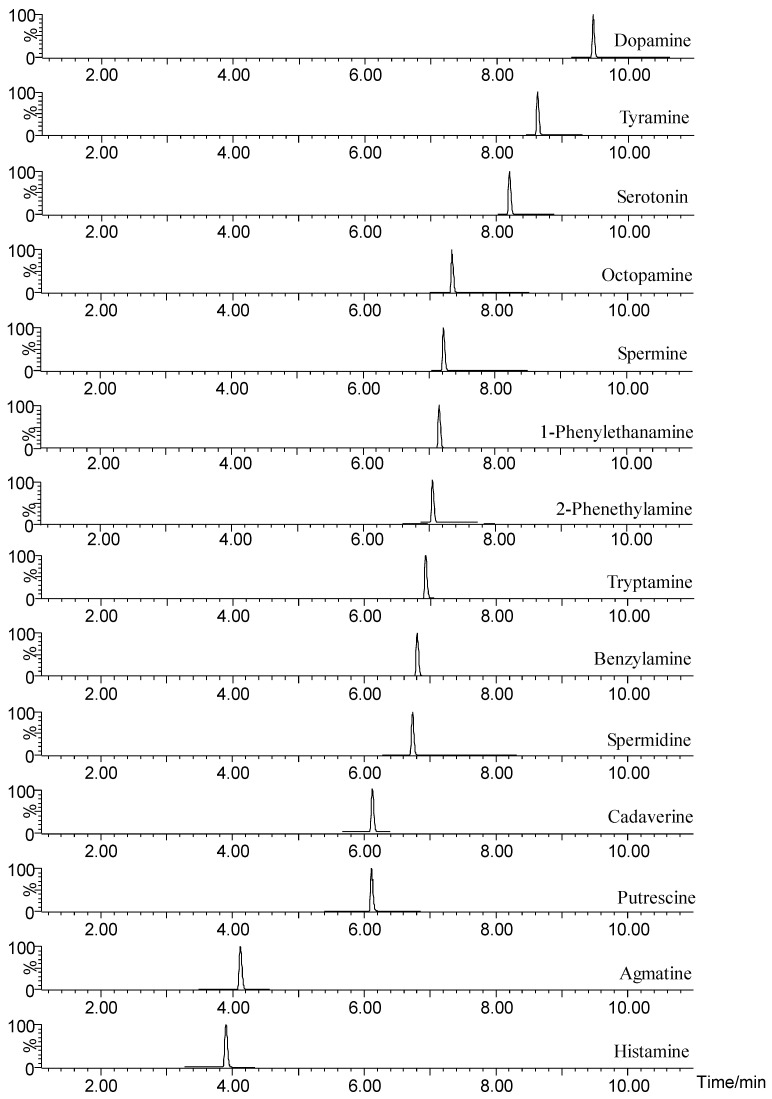
A chromatogram of 14 derivatized BAs in a 1000 nM standard mixture.

**Figure 3 molecules-28-00184-f003:**
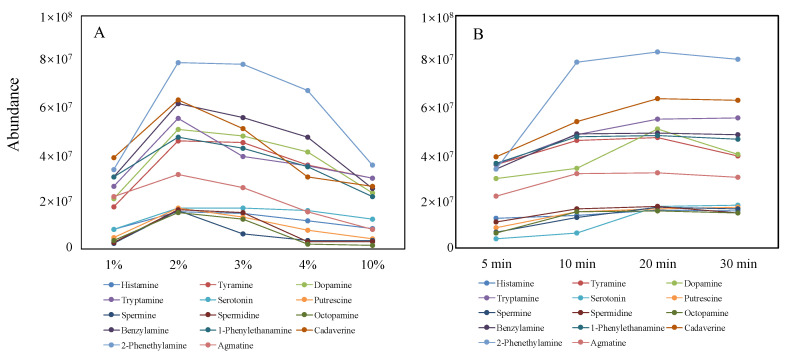
Optimization of the concentration of benzoyl chloride (**A**) and vortex time (**B**).

**Figure 4 molecules-28-00184-f004:**
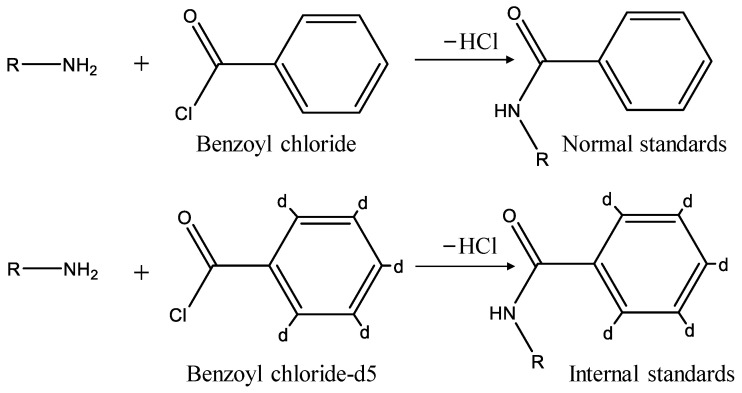
The derivatization scheme for BAs is derivatized by benzoyl chloride.

**Table 1 molecules-28-00184-t001:** MRM settings of the biogenic amine derivatives.

Name	Precursor Ions	Product Ions (1)	Collision Energy (eV)	Product Ions (2)	Collision Energy (eV)	Retention Time (min)
Histamine	216	105 *	40	77	40	3.90
Tyramine	346	105 *	20	77	45	8.61
Dopamine	466	105 *	30	77	40	9.52
Tryptamine	265	144	35	105 *	45	6.95
Serotonin	385	264	25	105 *	40	8.42
Putrescine	297	105 *	20	77	40	6.03
Spermine	619.6	497 *	20	105	35	7.21
Spermidine	458	162 *	20	105	40	6.79
Octopamine	344	105 *	25	77	40	7.40
Benzylamine	212.1	91 *	20	77	40	6.83
1-Phenylethanamine	226.2	122 *	25	105	40	7.15
Cadaverine	311.2	105 *	20	77	40	6.22
2-Phenethylamine	226.2	105 *	20	77	40	7.02
Agmatine	235	176	20	105 *	40	4.11

* Means the ions with corresponding collision energy for quantification.

**Table 2 molecules-28-00184-t002:** Linear range, R^2^, LOD, LOQ, and matrix effect of 14 BA derivatives.

Name	Linear Range (nmol/L)	LOD (nmol/L)	LOQ (nmol/L)	R^2^	Matrix Effect (%)
Histamine	10–10,000	8.72	28.77	0.9989	−1.36
Tyramine	2–8000	0.55	1.80	0.9988	2.71
Dopamine	8–10,000	4.36	14.39	0.9994	−4.87
Tryptamine	20–50,000	19.90	65.67	0.9998	−5.90
Serotonin	20–50,000	17.44	57.55	0.9996	−10.32
Putrescine	20–50,000	16.44	54.25	0.9995	−8.76
Spermine	2–8000	1.09	3.60	0.9998	−11.9
Spermidine	2–8000	2.18	7.19	0.9996	3.76
Octopamine	2–8000	0.44	1.44	0.9998	−8.16
Benzylamine	2–8000	0.27	0.90	0.9999	−14.21
1-Phenylethanamine	2–8000	0.13	0.44	0.9998	−6.20
Cadaverine	2–8000	0.14	0.45	0.9998	−9.04
2-Phenethylamine	2–8000	1.08	3.56	0.9994	−4.92
Agmatine	2–8000	1.09	3.60	0.9961	−6.37

**Table 3 molecules-28-00184-t003:** Recoveries and RSDs of 14 BA derivatives, including extraction processing, derivatization, and LC–MS analysis.

Name	Recoveries			RSDs		
	10 nM	50 nM	1000 nM	1 nM	50 nM	1000 nM
Histamine	103.7%	115.4%	92.1%	7.9%	8.9%	7.4%
Tyramine	102.2%	84.6%	88.2%	10.0%	5.4%	8.4%
Dopamine	105.1%	99.2%	92.6%	9.6%	3.3%	4.8%
Tryptamine	119.3%	88.5%	88.9%	9.3%	5.4%	3.0%
Serotonin	104.1%	86.3%	74.9%	8.4%	6.3%	4.5%
Putrescine	90.5%	88.1%	80.5%	8.7%	6.2%	3.9%
Spermine	99.7%	92.5%	81.2%	7.8%	6.7%	5.1%
Spermidine	100.5%	95.4%	79.2%	5.6%	5.3%	6.2%
Octopamine	97.6%	90.4%	77.2%	7.2%	9.3%	7.9%
Benzylamine	100.5%	104.5%	79.9%	5.4%	8.7%	9.3%
1-Phenylethanamine	99.8%	105.9%	108.1%	6.6%	9.6%	9.5%
Cadaverine	98.4%	101.2%	105.6%	6.0%	6.2%	9.4%
2-Phenethylamine	101.7%	92.0%	82.2%	7.1%	6.7%	8.3%
Agmatine	104.9%	99.2%	86.1%	5.9%	8.4%	8.4%

**Table 4 molecules-28-00184-t004:** The content comparison of biogenic amines in fish from a morning market and supermarket.

BAs (nmol/kg)	Mackerel		Tuna	
	Supermarket (n = 5)	Morning Market (n = 5)	Supermarket (n = 5)	Morning Market (n = 5)
Histamine	3087.21 ± 196.35	1872.73 ± 283.72	34.42 ± 8.3	10.76 ± 2.26
Tyramine	1499.73 ± 172.71	446.62 ± 53.66	43.82 ± 4.23	15.92 ± 4.12
Dopamine	-	-	-	-
Tryptamine	260.64 ± 12.59	134.34 ± 16.29	184.79 ± 2.07	186.07 ± 1.21
Serotonin	23.82 ± 0.82	22.87 ± 2.09	23.6 ± 0.66	24.21 ± 0.64
Putrescine	-	-	256.93 ± 61.95	59.07 ± 9.06
Spermine	-	-	5.52 ± 2.01	0 ± 0
Spermidine	-	-	70.56 ± 22.85	26.19 ± 13.36
Octopamine	-	-	-	-
Benzylamine	1.55 ± 0.77	1.16 ± 0.32	0.76 ± 0.36	0.9 ± 0.49
1-Phenylethanamine	8.37 ± 3.11	-	-	-
Cadaverine	106.49 ± 6.63	103.59 ± 8.01	2.07 ± 0.94	2.51 ± 0.85
2-Phenethylamine	1062.8 ± 112.35	813 ± 74.49	4.08 ± 2.5	3.09 ± 1.32
Agmatine	13.02 ± 1.28	16.17 ± 3.45	2.16 ± 0.72	8.48 ± 3.56

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
