# Peer review of "The Development of an Ultra-Performance Liquid Chromatography–Tandem Mass Spectrometry Method for Biogenic Amines in Fish Samples"

_molecules, 2022, doi:10.3390/molecules28010184_

Round 1

Reviewer 1 Report

The authors described a UPLC-MS/MS method and the application in detecting 14 biogenic amines in fish samples. The methods were optimized such as flow rate and gradient. This study is helpful for the need to analyze biogenic amines in food for the quality purposes. Please consider the comments as below.

1.      The molecular methods such as PCR methods can also detect biogenic amines, which should be mentioned in Introduction.

2.      Please add up-to-date references for LC detection methods in Introduction. For instance, line 43, ref 8-10, these references are from over 10 years ago, and can not reflect current technology “In recent years (line 41)”.

3.      Please introduce tandem mass spectrometry in Introduction.

4.      There is no description of Fig 1 in the text (line 79). How many BAs were tested? How was the separation?

5.      Fig 1 shows 9 benzoyl chloride derivatized BAs not 14 BAs in Table 1. Please discuss the inconsistency.

6.      Fig 2B, the x-axis is the same as Fig 2A and does not show 4 time points.

7.      Line 107, “It was obvious that…”

8.      It looks like 2% benzoyl chloride makes obvious difference from 4% in Fig 2A for all the BAs. Is there any explanation for this result?

9.      Please show the optimization data of flow rate and gradient. I only found the final decision on the best choice but did not see the comparison.

Reviewer 2 Report

The authors proposed a UPLC method to determined biogenic amine in fish samples. The manuscript is very poorly presented and organized. No sufficient results were shown and lack in novelty, therefore it’s not recommended for publication.

The manuscript is full of typing errors and the organization is all over the place. Some examples are followed:

42-43 The sensitivity depends on the analytes and the matrix complexity, so this sentence can induce errors.

69-70 The optimal collision energy (CE) was optimized for each product ion “table 1”. Should be “presented in table 1” or (table 1).

Not sure why table 1 only appears 2 pages later. Same happen with figure 1

It’s not clear which ion and/or energy used for the quantification

72 mM/L?

78  40 oC should be C – This happens all over the text.

Figure 1 nine benzoyl? The chromatogram is just an overlay chromatogram. The authors should provide a chromatogram with the standard mixture and the fish sample.

Figure 2 b x axis with wrong values. The authors may consider providing the derivatization reaction.

2.3 please explain why using more benzoyl chloride lower the reaction? With the data, I think the authors should have test 3% benzoyl chloride. How did the authors confirm the derivatization yield?

R2 is determination coefficient and why in table 2 show 8/9 digits of R2?

It’s not clear how the accuracy was determined.

The material of the column was not indicated

It’s not clear what part of the fish was used and also those 2 were chosen as samples

Reviewer 3 Report

This study reports the method development for measuring BAs in fish samples using LCMS. The findings could be important to the relevant fields. Nevertheless, there are quite a number of confusions and concerns for this manuscript. The specific comments are as follows:

1.       Suggest specifying in the Abstract which 14 BAs were measured in this study. Same for line 54.

2.       Line 15: The term “formic acid in acetonitrile (0.1%)” is rather confusing. It should be “acetonitrile with 0.1% formic acid” or something with a similar meaning.

3.       Line 33: What is the range of BAs content in food that can be used to indicate the food quality, especially in fish?

4.       Suggest providing the chemical structures of the 14 BAs measured in this study.

5.       Line 74: Should provide the data for the statement “mobile phase elution gradient and flow rate were optimized” for better evaluation by the readers.

6.       Line 120: Please elaborate in the manuscript how “internal standard-corrected response” was calculated.

7.       Line 123: Please specify the range that was reported by former studies.

8.       Line 132-133: Please elaborate on how to conclude that the matrix effect ranged from -20% to 20%. Not data were shown to support the claim.

9.       Line 136: What kind of standards (normal or internal) was spiked into the samples? What samples were used? It should be the blank matrix samples. Please specify in the manuscript.

10.   Line 137-140: No data were shown to support the claims on recoveries and RSD.

11.   Line 147: Please specify which “edible parts” of the fish were used. Was that the fresh included with the skin?

12.   Legend Figure 1: How many BAs are there actually?

13.   Figure 2B: this figure is incorrect.

14.   Please include the graph with data points for the linear range mentioned in Table 2.

15.   What kind of samples was used as the blank to evaluate the data of matrix effect in Table 2.

16.   Section 3.1 (line 164): Please provide the details for the internal standards (IS) of the 14 BAs.

17.   Line 169: Why was benzoyl chloride-d5 used in this study? If it was used to generate the deuterated IS for the 14 BAs, please provide data for the IS, for example spectra as in Figure 1. How are the IS data different from the normal standard?

18.   Line 175: Should show the data for the optimization of derivatization.

19.   What is meant by “calibration standards and internal standards mixture”?  What were the calibration and internal standards used?

20.   Line 181: What is meant by “separately”?

21.   Line 198: What is “acetonitrile, within 0.1% FA” meant?

22.   Line 199: Please describe the homogenization step.

23.   Section 3.2 & 3.3: Suggest preparing a flow chart for the steps of derivatization of standards and samples.

24.   What are the internal standards? They were not specified in the manuscript. The internal standards should be the heavy isotope-labeled BAs.

25.   Line 229: Calibration curve was prepared using normal or deuterated standards? What is the internal standard meant here?

26.   Line 235: Please describe what is the “blank fish sample”. How was it obtained and prepared?  

27.   Line 238: Why was breast milk used in this study?

28.   Conclusion section: The authors claimed that the formic acid extraction reagent was friendly and cost-effective. Did they perform the extraction without acid or with other acids for such a claim? In addition, they claimed that benzoyl chloride was more rapid and simple. Did they perform the derivatization with other reagents for such a claim? Please provide the data for the claims.

Others:

29.   What is an “EP” tube?

30.   What is meant by “mM/L”?

31.   The use of the abbreviation is not consistent throughout the text.

32.   Please be consistent in using capitalization for the chemical names.

33.   There are many formatting errors found in the manuscript. 

Round 2

Reviewer 3 Report

The manuscript has improved a lot after the revision. Minor comments are as follows:

1.     Please describe the “blank fish sample” (comment 26) in the manuscript rather than just answered in the response to reviewer.

Others:

2.     Line 141: Should not be “four”.

3.     Line 267: Take note on the chemical formula